# MRI Assessed Placental Location as a Diagnostic Tool of Placental Invasiveness and Maternal Peripartum Morbidity

**DOI:** 10.3390/diagnostics14090925

**Published:** 2024-04-29

**Authors:** Charis Bourgioti, Marianna Konidari, Makarios Eleftheriades, Marianna Theodora, Maria Evangelia Nikolaidou, Konstantina Zafeiropoulou, Chara Tzavara, Stavros Fotopoulos, George Daskalakis, Lia Angela Moulopoulos

**Affiliations:** 1First Department of Radiology, School of Medicine, National and Kapodistrian University of Athens, Aretaieion Hospital, 11528 Athens, Greece; marianna_konidari@hotmail.com (M.K.); kon.zafeiropoulou@gmail.com (K.Z.); 2Second Department of Obstetrics and Gynaecology, School of Medicine, National and Kapodistrian University of Athens, Aretaieion Hospital, 11528 Athens, Greece; makarios@hotmail.co.uk; 3First Department of Gynaecology and Obstetrics, School of Medicine, National and Kapodistrian University of Athens, Alexandra Hospital, 11528 Athens, Greece; martheodr@gmail.com (M.T.); gdaskalakis@yahoo.com (G.D.); 4Department of Gynaecology and Obstetrics, IASO Maternity Hospital, 15123 Athens, Greece; marilianikolaidou@gmail.com (M.E.N.); fotostav@yahoo.gr (S.F.); 5Department of Health, Epidemiology and Medical Statistics School of Medicine, National and Kapodistrian University of Athens, 11527 Athens, Greece; htzavara@med.uoa.gr

**Keywords:** placenta previa, placenta accreta spectrum (PAS), maternal outcome, MRI

## Abstract

Placenta accreta spectrum (PAS) disorder is one of the leading causes of peripartum maternal morbidity and mortality; its early identification during pregnancy is of utmost importance to ensure the optimal clinical outcome. The aim of the present study is to investigate the possible association of the presence and type/location of placenta previa on MRI with PAS and maternal peripartum outcome. One hundred eighty-nine pregnant women (mean age: 35 years; mean gestational age: 32 weeks) at high risk for PAS underwent a dedicated placental MRI. All women underwent a C-section within 6 weeks from the MRI. All MRIs were evaluated by two experienced genitourinary radiologists for presence, type (complete/partial vs. marginal/low lying), and location (anterior vs. anterior-posterior vs. posterior) of placenta previa. Statistical analysis was performed for possible association of type/location of previa with placental invasiveness and peripartum outcomes. Intraoperative information was used as a reference standard. Complete/partial previa was detected in 143/189 (75.6%) and marginal/low lying previa in 33/189 (17.5%) women; in 88/189 (46.6%) women, the placenta had anterior–posterior, in 54/189 (28.6%) anterior and in 41/189 (21.7%) posterior. Complete/partial previa had an at least 3-fold probability of invasiveness and was more frequently associated with unfavorable peripartum events, including massive intraoperative blood loss or hysterectomy, compared to low-lying/marginal placenta. Posterior placental location was significantly associated with lower rates of PAS and better clinical outcomes. In conclusion, the type and location of placenta previa shown with MRI seems to be associated with severity of complications during delivery and should be carefully studied.

## 1. Introduction

Typically, placental implantation occurs at the uterine fundus due to the highest endometrial flow, and less frequently at the anterior and posterior wall. Placental implantation at the lower uterine segment, either covering (placenta previa) or reaching within 2 cm from the internal cervical os (low-lying placenta), is estimated to occur in 1 in 200 pregnancies at pregnancy term, and its incidence appears to have risen following the increasing rate of cesarean deliveries [1]. Changes of myometrial contractility and disrupted contraction waves in the endometrium due to a previous cesarean section may account for this lower placental implantation [2].

Placenta accreta spectrum (PAS) disorder is a major obstetrical complication associated with considerable maternal and fetal morbidity or even mortality, especially if it remains undiagnosed prenatally. The incidence of PAS is significantly higher in women with placenta previa and history of cesarean deliveries; interestingly, in cases of placenta previa and low-lying placenta, the risk of PAS was found to be 3%, 11%, 40%, 61%, and 67% with a history of one, two, three, four, and five or more prior cesarean deliveries, respectively [3]. The combination of placenta previa and history of prior uterine instrumentation was associated with high rates of PAS development, reaching 50–67% [4].

Early identification of abnormal placental location and invasiveness is of great importance to surgeons due to the increased risk of antepartum hemorrhage at the time of delivery, as it provides more adequate preoperative planning and better patient counseling [1,2,3,4,5]. The Color Doppler Ultrasound technique (CDUS) is the primary diagnostic tool for evaluating abnormal placentation, with reported sensitivity and specificity values up to 90.7% and 96.9%, respectively [6]. Magnetic Resonance Imaging (MRI) may also serve as a reliable alternative to the US modality for the identification of PAS, with reported overall sensitivity and specificity values ranging from 75 to 100% and from 65 to 100% [7,8].

Particularly, the appropriate imaging modality for detecting placenta previa is transvaginal ultrasonography; MRI is a complementary, non-invasive diagnostic tool which can confirm ultrasound findings. It may be helpful in sonographically ambiguous cases, as it covers a larger field of view and is a more reproducible imaging technique [9,10].

The aim of this study was to investigate any association of the presence and type of placenta previa on MRI with PAS and maternal peripartum outcome.

## 2. Materials and Methods

### 2.1. Patient Selection

The institutional review board approved this prospective study, and a written informed consent was obtained from all participants (ethics registration number B-196/13.10.2016). 

Between March 2016 and March 2021, 197 women in the third trimester of pregnancy were referred for a dedicated placental MRI due to abnormal placenta location (i.e., placenta covering or located within 2 cm from the internal cervical os) and/or suspicious findings of PAS at the second-trimester ultrasound. Eight women did not proceed with the MRI because of claustrophobia (*n* = 5) or body habitus (*n* = 3). One hundred eighty-nine women completed the MRI and formed our study group.

### 2.2. MRI Protocol

All study participants underwent placental MRI at 1.5 T (*n* = 91) or 3.0 T (*n* = 98) units. 

The acquisition protocol for 1.5 T MRI included: T2-weighted single-shot turbo spin-echo sequences in the axial, sagittal, and coronal planes (repetition time msec/echo time msec, 510–568/80); T2-weighted turbo spin-echo sequence in all three planes (6300/90) and, when extrauterine spread was suspected, parallel and perpendicular to the cervical canal (2500/90); axial T1-weighted turbo spin-echo fat-suppressed sequence (730/6.9). The acquisition protocol for 3.0 T imaging included: T2-weighted images in all three planes (4345–4666/100); T2-weighted turbo spin-echo sequence perpendicular and parallel to the cervix (4807/90), when extrauterine spread was suspected, and axial T1-weighted turbo spin-echo fat-suppressed sequence (723/8.0). Intravenous paramagnetic contrast was not administered, due to fetal safety considerations. 

Detailed technical parameters of the applied acquisition protocols for both 1.5 T and 3.0 T MRIs are presented in Appendix A.

### 2.3. Image Interpretation

All prospectively collected MRI data were secondarily evaluated with consensus by two experienced genitourinary radiologists (Reader 1: 25 years of experience; Reader 2: 11 years of experience) for presence and type of placenta previa. In the present study, we used the initial clinical classification of placenta previa (complete, partial, marginal and, low-lying), depending on the extent of internal cervical os coverage by the placenta [11]. Complete previa was assigned when the placenta completely covered the internal os; partial previa when the placenta partially covered the internal os; marginal previa when the placental edge reached the margin of the internal cervical os and low-lying placenta when the placenta reached within 2 cm from the internal cervical os (Figure 1). Since in the last few years, all placentas overlying the internal cervical os, to any degree, are defined as previas and those near to but not overlying the os, as low-lying, we decided to group complete with partial previas and marginal with low-lying types, to facilitate statistical analysis [1,11].

Placental implantation relative to the uterine wall was recorded as well. Anterior placental location was assigned when the placenta was attached to the anterior uterine wall; posterior placental location, when the placenta was attached to the posterior uterine wall; anterior-posterior location was assigned when the placenta extended to both anterior and posterior uterine walls and lateral location when the placenta was predominantly attached to any of the lateral sides of the uterus (Figure 2, Figure 3 and Figure 4).

### 2.4. Maternal Intraoperative Course

All patients underwent a cesarean section within six weeks from the MRI study. 

The following clinical information related to the intraoperative maternal outcome was recorded in detail: onset of delivery (emergency vs. scheduled), duration of delivery (min), intraoperative blood loss (estimates were based on transfusion data, weighing of bloody dressings, and visual inspection of blood on objects in the operating room that cannot be weighed, e.g., floor), need for massive transfusion (>2000 mL, given that in an uncomplicated cesarean section, blood loss is usually <1000 mL), hysterectomy treatment, need for bladder repair (i.e., minimal, when there was no clear surgical plane between uterus and bladder but no or minimal injury of the bladder wall occurred during vesicouterine fold detachment; extensive, when cystotomy or partial cystectomy was required) and Intensive Care Unit (ICU) hospitalization.

### 2.5. Standard of Reference

Intraoperative information (surgical evidence) was used as the standard of reference for placental invasiveness and extrauterine placental spread, according to FIGO (Federation International of Obstetrics and Gynecology) definitions [12]. Histologic examination of the uteroplacental specimen (following hysterectomy) or gross inspection of the detached placenta (following conservative treatment) was also performed to assist surgical evidence (pathologic evidence).

### 2.6. Statistical Analysis

Quantitative variables were expressed as mean values (Standard Deviation) and as median (Interquartile Range), while qualitative variables were expressed as absolute and relative frequencies. For the comparison of proportions, chi-square and Fisher’s exact tests were used. Kruskall–Wallis test was used for the comparison of continuous variables among more than two groups. Bonferroni correction was used to control for type I error. Logistic and linear regression analyses were used to find if the type of previa and location were significantly associated with all under study outcomes, after adjusting for age, prior uterine surgeries, maternal comorbidities, gestational age at delivery, number of IVFs, number of cesarean sections and number of fetuses. Adjusted odds ratios (OR) with 95% confidence intervals (95% CI) were computed from the results of the logistic regression analyses. Adjusted regression coefficients (β) with standard errors (SE) were computed from the results of the linear regression analyses. All reported *p* values are two-tailed. Statistical significance was set at *p* < 0.05 and analyses were conducted using SPSS statistical software (version 22.0).

## 3. Results

### 3.1. Study Group

The sample consisted of 189 pregnant women (mean age 34.9 years, SD = 4.9 years; mean gestational age 35.3 weeks, SD = 2.1 weeks). A history of cesarean section or any other uterine surgery except c-section was recorded in 129/189 (68.2%) and 82/189 (43.4%) of the participants, respectively. Placenta previa was detected in 143/189 (75.7%) gravid women. 

In 88/189 (46.6%) pregnant women the placenta had an anterior–posterior location; in 54/189 (28.6%) women the placenta was located anteriorly and in 41/189 (21.7%) placenta was located posteriorly. In 6/189 (3.2%) women the placenta was attached predominantly at the lateral side of the uterus; to facilitate statistical analysis, these six patients were excluded from further evaluation.

One hundred fifty-three to 189 study participants (81.0%) were diagnosed with PAS; placenta percreta was found in 83/189 (43.9%) gravid women; bladder involvement in 65/189 (34.4%) and parametrial involvement in 34/189 (18%) women. Adverse peripartum outcomes, including need for massive transfusion during delivery, hysterectomy and minimal or major bladder repair were recorded in 32/189 (17.4%), 69/169 (36.5%) and 65/189 (34.4%) women, respectively. After the delivery, most participants required ICU monitoring (181/189, 95.8%). Median blood loss during surgery was 750 mL (IQR: 125–1625), median duration of delivery was 75 min (IQR: 60–120 min) and median duration of ICU stay was 1 day (IQR:0–1). The sample’s sociodemographic and placenta related characteristics are fully presented in Table 1 and Table 2, respectively.

### 3.2. Association between Previa Type and Clinical Outcomes

The incidence of PAS was significantly higher in women with complete/partial compared to those with low-lying/marginal placenta (*p* = 0.004); bladder and parametrial involvement were more frequent in women with complete/partial compared to low-lying/marginal previa (*p* = 0.002 and *p* = 0.006, respectively). Participants with complete/partial previa, experienced greater blood loss during surgery compared to women with low-lying/marginal previa (*p* < 0.001). Additionally, women with complete/partial previa were treated with hysterectomy more frequently compared to those with low-lying/marginal previa (*p* = 0.022). Prolonged operation times and need for ICU hospitalization were more frequently recorded in cases with complete/partial compared to low-lying/marginal previa cases (*p* = 0.004 and *p* < 0.001, respectively). Association between previa type and clinical outcomes is presented in detail in Table 3.

### 3.3. Association between Placental Location and Clinical Outcomes

Almost all peripartum outcomes differed significantly with varying placental location (Table 4). After Bonferroni correction, it was found that the incidence of PAS was significantly lower in posterior compared to the anterior and anterior/posterior placentas (*p* = 0.001 and *p* < 0.001 respectively). Moreover, percreta was significantly less common in posterior compared to anterior and anterior/posterior placentas (*p* = 0.001 and *p* < 0.001 respectively). Placental extrauterine spread and bladder involvement was significantly more frequent in anterior/posterior compared to anterior (*p* = 0.011, for both outcomes) and to posterior placentas (*p* < 0.001, for both outcomes). The frequency of parametrial involvement was significantly higher in anterior/posterior compared to posterior placental location (*p* = 0.001). Blood loss during surgery was significantly greater in anterior/posterior compared to anterior (*p* = 0.004) and to posterior placentas (*p* < 0.001). Hysterectomy treatment was more frequent in anterior/posterior compared to anterior (*p* = 0.012) and to posterior placenta location (*p* < 0.001); also, need for hysterectomy was more frequent in anteriorly compared to posteriorly located placentas (*p* = 0.001). The need for any bladder repair was significantly greater in anterior/posterior placenta compared to anterior (*p* = 0.011) and to posterior placentas (*p* < 0.001). Surgical time and ICU stay were more prolonged in women with anterior/posterior compared to anterior (*p* = 0.005 and *p* = 0.010 respectively) and posterior placentas (*p* < 0.001 and *p* < 0.001 respectively).

### 3.4. Multiple Regression Analysis Results with All under Study Outcomes as Dependent Variables and Previa Type as Independent Variable

After adjusting for age, gestational age at delivery, prior uterine surgeries, maternal comorbidities, number of cesarean sections, number of IVFs, and number of fetuses, it was found that in complete/partial previa cases the probability of PAS was 3.32 times higher than in low-lying/marginal previa cases. Moreover, in complete/partial previa group the probability of extrauterine spread and bladder involvement was 4.06 and 4.33 times higher, respectively compared to the low-lying/marginal previa group. Additionally, in complete/partial previa cases the probabilities of parametrial invasion, hysterectomy treatment and bladder repair were 8.88, 3.73 and 4.33 times higher than in the low-lying/marginal group (Figure 4). Delivery and ICU hospitalization times were significantly greater in women with complete/partial previa compared to those with low-lying/marginal previa. The above results are presented in detail in Table 5.

### 3.5. Multiple Regression Analysis Results with All under Study Outcomes as Dependent Variables and Location as Independent Variable

After adjusting for age, gestational age at delivery, prior uterine surgeries, maternal comorbidities, number of c-sections, number of IVFs, and number of fetuses (Table 6) it was found that the probability of PAS in anteriorly and anteriorly/posteriorly located placentas was 6.94 and 7.38 times higher respectively compared to posteriorly located placentas. Furthermore, the probability of less aggressive forms of invasiveness (accreta/increta) was lower in anterior and anterior-posterior PAS than that of posterior invasive placenta by 89% and 94%, respectively.

Additionally, in anterior and anterior/posterior placentas, the probability of extrauterine placental spread was 19.27 and 49.63 times higher, respectively compared to posteriorly located placentas. The probability of placental extension to parametrial tissues was 12.12 times higher in anterior/posterior than in posterior placenta cases. Women with anterior and anterior/posterior placenta had an increased probability (16.32- and 55.49-times, respectively) to be treated with hysterectomy compared to women with posterior placentas. Anterior/posterior placentas were associated with significantly greater blood loss during surgery, increased duration of delivery and ICU hospitalization, compared to posterior placentas. Interestingly, the probability of acute onset of labor in anterior/posterior placenta was 76% lower than in posteriorly located placentas.

## 4. Discussion

In the present study, we investigated the association of placental location (i.e., complete/partial vs. low-lying/marginal previa) and position (i.e., anterior/posterior vs. anterior vs. posterior) on MRI with placental invasiveness and maternal outcome during delivery, in a large population of gravid women. Our results showed that complete/partial previa had at a least 3-fold probability of placental invasiveness and, it was more frequently associated with unfavorable peripartum events, including massive intraoperative blood loss or hysterectomy compared to low-lying/marginal placenta. Posterior placental location was significantly associated with lower rates of PAS and better clinical outcomes compared to anterior or anterior/posterior placenta.

Abnormalities in placental location commonly known as placental implantation anomalies, include low-lying placenta, placenta previa and placenta accreta spectrum (PAS) disorders [2]. The etiology of these abnormalities is still under investigation although their incidence has been increasing during the last decades, possibly due to the rising number of cesarean sections. Other predisposing factors that increase the possibility of such placental implantation anomalies include prior uterine instrumentations (e.g., polypectomy, hysteroscopy, curettage), advanced maternal age, multiparity and in vitro fertilization [2]. 

Placental relation to the internal cervical os (previa/low-lying) and placental attachment to the uterine wall (anterior, posterior or anterior/posterior) are routinely evaluated on screening transabdominal ultrasound, performed between 18 and 22 weeks of gestation. According to the Royal College of Obstetricians and Gynecologists recommendations, the term ‘placenta previa’ is used when the placenta lies directly over the internal os; for gestational ages greater than 16 weeks, the placenta is considered ‘low-lying’ when the placental edge is less than 20 mm from the internal os, and normal when the placental edge is 20 mm or more away from the internal os on ultrasound examination. In a significant number of cases, changes in the lower uterine segment during the third trimester of pregnancy result in placental ‘migration’ and resolution of low-lying placenta. A process termed throphotropism is responsible for this “migration” of the placenta; the placenta tends to grow toward the compartment of the uterus with the best vascular supply (typically the fundus), while the placental portion near to the cervix regresses, due to its poorer vascular supply. However, this is less likely to occur in women with a previous caesarean delivery. A follow-up transabdominal ultrasound, at approximately 32 weeks of gestation, is useful to confirm the exact placental location. In general, it is recommended that women with a history of previous caesarean section with anterior low-lying placenta or placenta previa at routine second trimester ultrasound scan should undergo investigation for placenta accreta spectrum [13]. 

When there is suspicion for placenta previa or marginal/low-lying placenta, or in cases of technical difficulties obscuring placental borders (i.e., overdistended maternal bladder causing stretching of the cervical canal, localized uterine contractions mimicking placental tissue or position of the fetal head across the area of the internal os), a transvaginal approach can be performed for more accurate diagnosis. However, transvaginal imaging should be undertaken with care in advanced pregnancies, since there are theoretical concerns about increasing the risk of bleeding or causing premature rupture of membranes or infection, when the membranes have already ruptured [5,13,14]. 

Placental implantation anomalies predispose a higher risk of adverse maternal outcome and particularly to hemorrhage during delivery compared to normal placental location. Previa location is considered an independent risk factor for placental invasiveness and pregnant women with placenta previa, even without PAS, are at an approximately 10-fold increased risk of postpartum vaginal bleeding [15]. Additionally, there is some evidence concerning potential association of placenta previa with poor perinatal outcome including preterm birth, small for gestational age (SGA) fetus or stillbirth. Therefore, accurate diagnosis of placental location and position is important for delivery planning [16].

In our study we found that type of previa, a feature easily recognized on MR images even from non-expert radiologists, had a significant impact on obstetric outcomes; complete/partial previa had an at least 3-fold probability of placental invasiveness and, major complications during delivery, including need for massive transfusion due to intraoperative blood loss or hysterectomy, compared to low-lying/marginal placenta. Since, in several studies, in accordance with our results, it has been reported that unfavorable peripartum events such as peri-or postpartum hemorrhage, increased maternal morbidity or even preterm delivery may be significantly more prevalent in women with complete placenta previa, it is of upmost importance to distinguishing between different types of placenta previa in order to optimize patient care [4,16,17]. In our study, unlike previously published reports, we have used MRI to identify the type of placenta previa, an imaging modality which allows for a quick non-invasive and accurate identification of placental position with reported diagnostic accuracy >90% [7]. Sagittal MR images optimally display the relationship between the placenta and the internal os and MRI may be used as a reliable alternative imaging modality to confirm placental location in cases of indeterminate sonographic findings, since it provides larger field-of-view, and is more reproducible than ultrasound. In our series, MRI demonstrated 100% detection rate for placental location and position based on intraoperative information. 

In accordance with the published literature, our results showed that posterior placental location was associated with lower rates of PAS and better clinical outcomes compared to those placentas which involve the anterior uterine wall [18]. A possible explanation for this is that the posterior myometrium is thicker, therefore, depth of placental invasion is usually limited, rarely invading its entire thickness. Interestingly, a recent clinical study reported that 78% of posterior PAS cases were placenta accreta type while, in addition, in a cohort study the estimated risk of posterior placenta invasion in general population was low, with a reported incidence about 1%. [18,19]. Although less frequent and aggressive, posterior PAS can also be associated with significant maternal morbidity especially if remains undiagnosed since it may be associated with peculiar vascular supply (i.e., from the anterior rectal artery that follows the shape of the curve of Douglas pouch), which potentially makes hemostasis control more challenging in these women, especially when obstetricians are less experienced. The performance of ultrasound to identify cases of posterior PAS is limited since there is technical difficulty in assessing the posterior myometrium as there is no urinary bladder contrast. According to FIGO MRIs recommended for the identification of posterior PAS compared to ultrasound as it allows better visualization of the posterior uterine wall, with reported accuracy values up to 73.5% and 52.4% respectively [13,18].

In general, MRI exhibits high diagnostic performance for the identification of PAS and particularly for the detection of extrauterine placental spread [7,20,21]. Previously published data extracted from our study group, showed MRI sensitivity and specificity for PAS equal to 95.7% and 81.8% for 1.5-T and 93.8% and 81.8% for 3.0-T, respectively. Additionally, sensitivity and specificity values for MRI detection of extrauterine placental spread were 100% and 96.7% for 1.5-T and 97% and 96.6% for 3.0-T, respectively; no statistically significant differences were found between 1,5-T and 3.0-T groups [22].

However, it is difficult to perform MRI in all patients at high risk for PAS, despite its added value to PAS diagnosis, especially in median to low- income countries due to cost effectiveness issues and limited availability. In our tertiary center we perform MRI in all patients with complete previa on second trimester ultrasound also taking into account any relevant clinical setting (i.e., history of one or more c-sections, in vitro fertilization, multiple uterine interventions, or history of previous PAS pregnancy), even in cases with normal sonographic findings. 

This study has some limitations. Firstly, our study sample was already at high risk for PAS; therefore, results may not be the same when applied to the general population. However, MRI is not a routine assessment tool for the evaluation of the pregnant population and its usage is justified when there is a clinical benefit from the performance of the study. Experienced radiologists in genitourinary radiology and obstetric imaging conducted this study; external validation of the results from general radiologists is probably needed to confirm reproducibility. Finally, management of patients at high risk for PAS disorder varies among different clinical centers and strongly depends on surgeons’ experience, patient’s desire for fertility preservation and availability of a multidisciplinary team, All the above parameters may affect peripartum outcome, including overall surgical time, adequate hemostasis, or uterine sparing vs. hysterectomy. 

In conclusion, we found that complete/partial placenta previa is associated with an increased incidence of PAS and adverse peripartum events compared to low-lying/marginal previa. In addition, anterior–posterior placental location is associated with higher incidence of placental invasiveness and with less favorable maternal outcome compared to both anterior or posterior placental location. MRI facilitates accurate prenatal identification of such placental morphology enhancing clinical suspicion of a more complicated pregnancy and offering clinicians a better chance to design delivery more carefully and improve therapeutic outcome and patient prognosis.

## Figures and Tables

**Figure 1 diagnostics-14-00925-f001:**
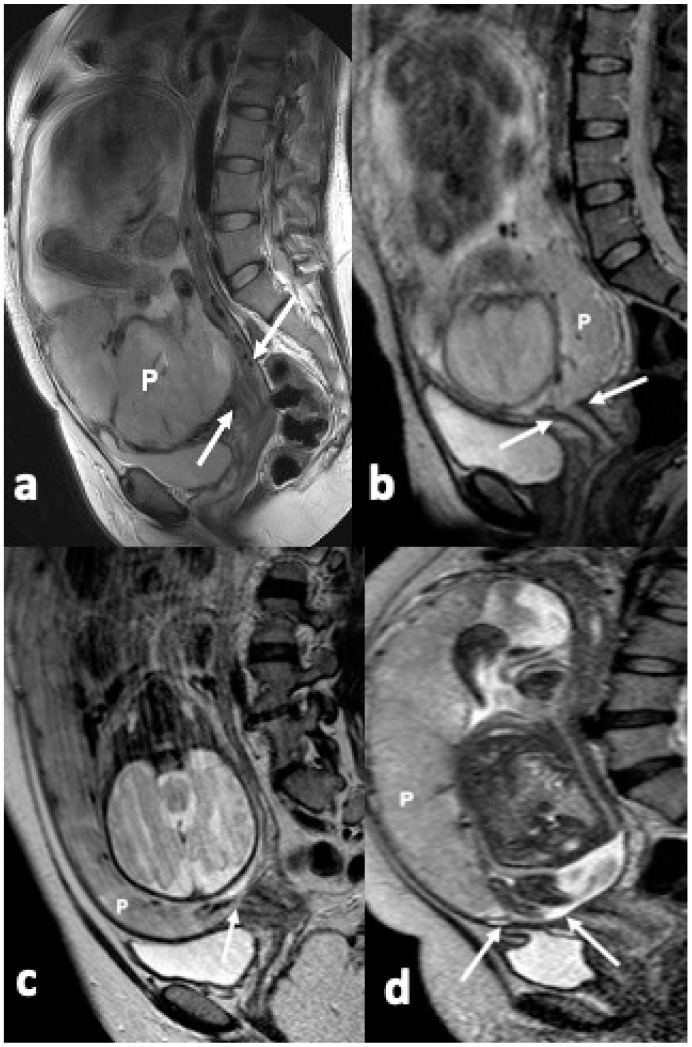
Previa types on MRI. T2W images in the sagittal plane demonstrate (**a**) complete (**b**) partial (**c**) marginal and (**d**) low lying previa. Note the association of the placenta (P) with the internal cervical os (arrows) in each case.

**Figure 2 diagnostics-14-00925-f002:**
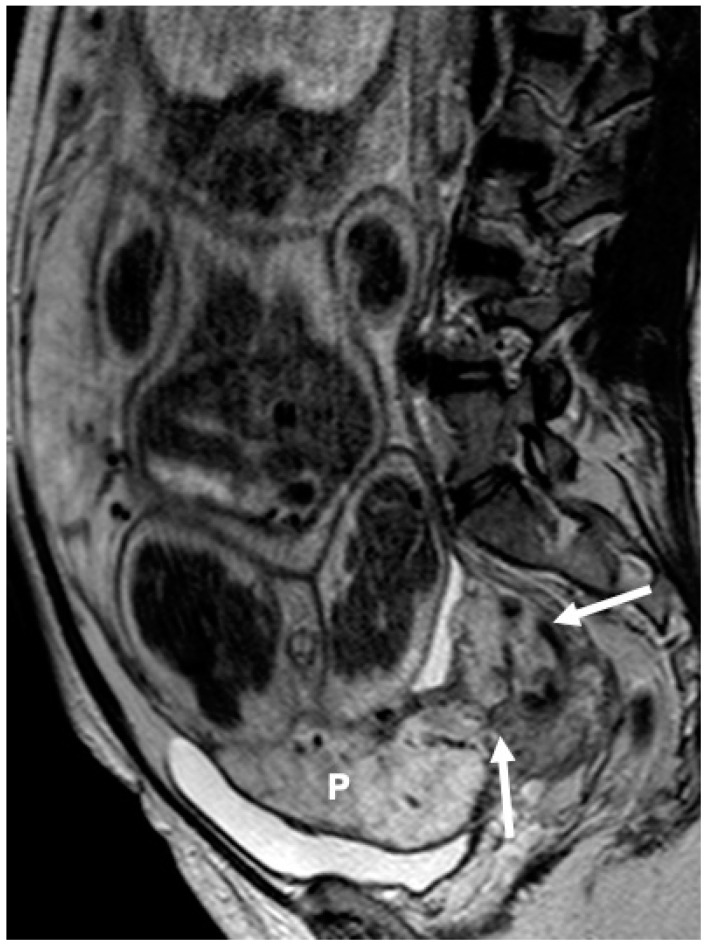
A 35-year-old woman at gestational week 34. Sagittal T2W (1.5 T) image shows an anteriorly located complete previa (P) with protrusion of placental tissue within the cervical canal (arrows). Placenta percreta was found intraoperatively. The patient was treated with hysterectomy required massive blood transfusion (>2500 mL).

**Figure 3 diagnostics-14-00925-f003:**
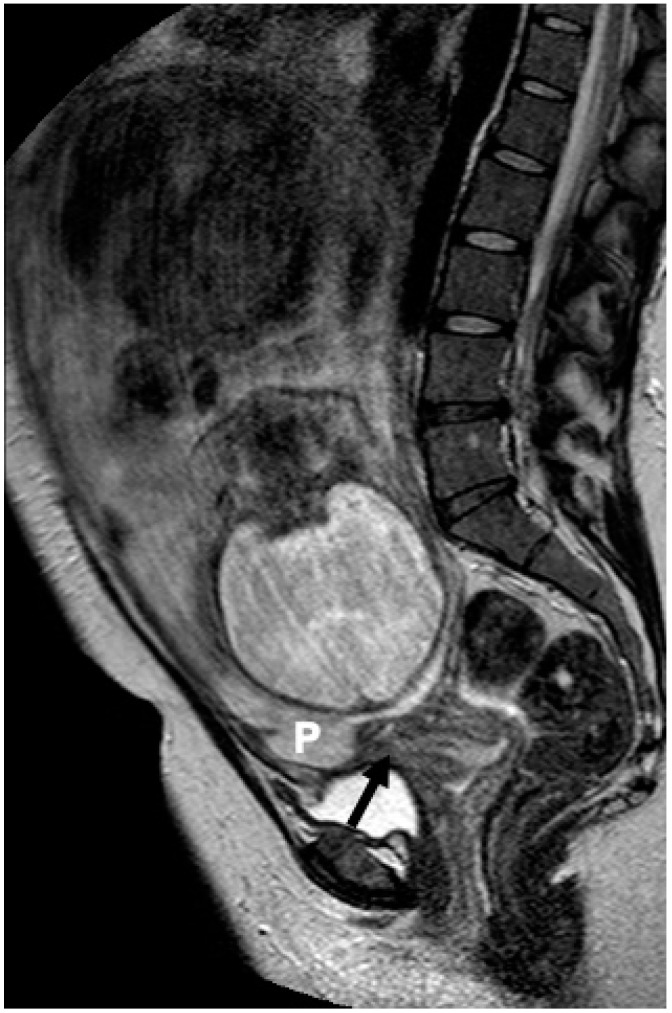
A 30-year-old woman at gestational week 32. Sagittal T2W image (1.5 T) shows an anteriorly located placenta (P) coming close to the internal cervical os (arrow). Normal placenta was diagnosed at the delivery.

**Figure 4 diagnostics-14-00925-f004:**
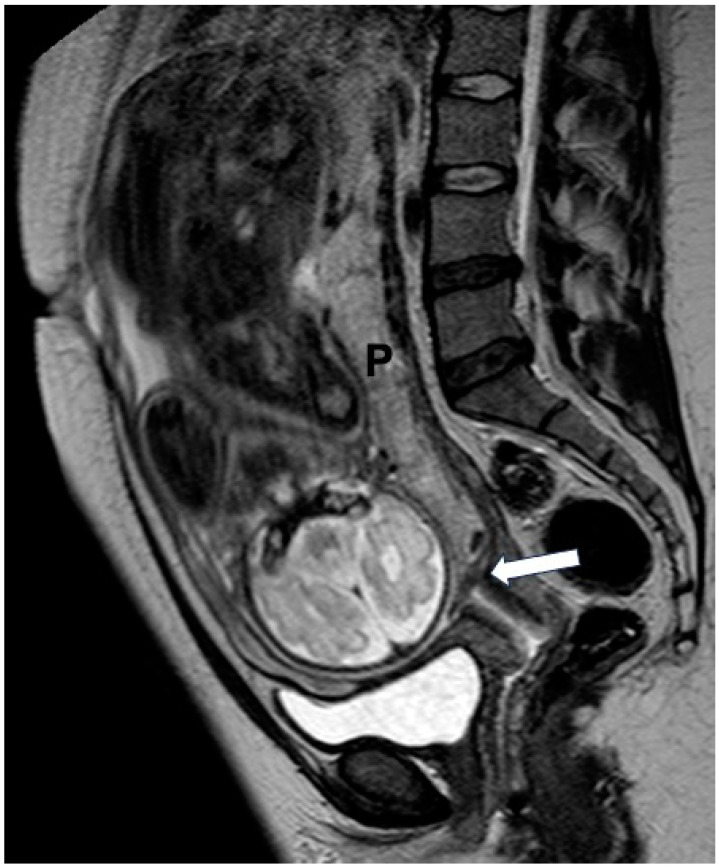
A 33-year-old-woman at gestational week 32 with history of leukemia. Sagittal T2W image (1.5 T) shows a posteriorly located placenta (P) partially covering the internal cervical os (arrow). There was no evidence of PAS during delivery.

**Table 1 diagnostics-14-00925-t001:** Sociodemographic and medical characteristics.

	*N* (%)
Age (years), mean (SD)	34.9 (4.9)
Prior uterine surgeries (any, except c-section)	82 (43.4)
Maternal comorbidities	45 (24.1)
Gestational age at delivery (weeks), mean (SD)	35.3 (2.1)
History of c-section Number of c-section, median (IQR)	129 (68.2) 1 (0–2)
Number of IVF	
0	167 (88.4)
1	18 (9.5)
2	4 (2.1)
Number of fetuses	
1	180 (95.2)
2	9 (4.8)

**Table 2 diagnostics-14-00925-t002:** Placenta-related findings.

	*N* (%)
Previa (type, any)	176/189 (93.1)
complete	115 (60.8)
low-lying	11 (5.8)
marginal	22 (11.6)
partial	28 (14.8)
Previa (type, groups)	176/189 (93.1)
complete/partial	143 (75.7)
low-lying/marginal	33 (17.5)
Placenta location	
anterior	54 (28.6)
anterior–posterior	88 (46.6)
lateral	6 (3.2)
posterior	41 (21.7)
Placenta location (groups)	
anterior	54 (29.5)
anterior–posterior	88 (48.1)
posterior	41 (22.4)
PAS (histology/intraoperative)	153 (81.0)
Degree of invasiveness (histology/intraoperative)	
absent	36 (19.0)
percreta	83 (43.9)
accreta/increta	70 (37.0)
Extrauterine spread (intraoperative)	66 (34.9)
Bladder invasion (intraoperative)	65 (34.4)
Parametrial invasion (intraoperative)	34 (18.0)
Blood loss during surgery (mL), median (IQR)	750 (125–1625)
Need for massive transfusion	32 (17.4)
Hysterectomy	69 (36.5)
Bladder repair	65 (34.4)
Duration of delivery (min), median (IQR)	75 (60–120)
Acute onset of labor	29 (15.3)
Need for IUC stay	181 (95.8)
Need for IUC stay (days), median (IQR)	1 (0–1)

Abbreviations: PAS, placenta accrete spectrum; IQR, interquartile range; c-section, caesarian section; ICU, intensive care unit.

**Table 3 diagnostics-14-00925-t003:** Participants’ outcomes associated with type of previa.

		Previa			*p*
		No	Complete/Partial	Low-Lying/ Marginal	
		*N* (%)	*N* (%)	*N* (%)	
PAS (intraoperative/histology)	No	1 (7.7)	22 (15.4)	13 (39.4)	0.004 ^+^
	Yes	12 (92.3)	121 (84.6)	20 (60.6)	
Degree of invasiveness (intraoperative/histology)	No	1 (7.7)	22 (15.4)	13 (39.4)	0.013 ^++^
	Percreta	4 (30.8)	69 (48.3)	10 (30.3)	
	Accreta/increta	8 (61.5)	52 (36.4)	10 (30.3)	
Extrauterine spread (intraoperative)	No	12 (92.3)	83 (58.0)	28 (84.8)	0.001 ^+^
	Yes	1 (7.7)	60 (42.0)	5 (15.2)	
Bladder involvement (intraoperative)	No	12 (92.3)	84 (58.7)	28 (84.8)	0.002 ^+^
	Yes	1 (7.7)	59 (41.3)	5 (15.2)	
Parametrial involvement (intraoperative)	No	13 (100.0)	110 (76.9)	32 (97.0)	0.006 ^+^
	Yes	0 (0.0)	33 (23.1)	1 (3.0)	
Blood loss during surgery, median (IQR)		250 (0–500)	1000 (250–2000)	250 (0–500)	<0.001 ^‡^
Need for massive transfusion (>2000 mL)	No	12 (92.3)	108 (78.3)	32 (97)	0.025 ^+^
	Yes	1 (7.7)	30 (21.7)	1 (3)	
Hysterectomy	No	10 (76.9)	83 (58)	27 (81.8)	0.022 ^+^
	Yes	3 (23.1)	60 (42)	6 (18.2)	
Bladder repair	No	12 (92.3)	84 (58.7)	28 (84.8)	0.002^+^
	Yes	1 (7.7)	59 (41.3)	5 (15.2)	
Duration of delivery, median (IQR)		60 (50–90)	80 (60–120)	60 (45–80)	0.004 ^‡^
Acute onset of labor	No	13 (100.0)	118 (82.5)	29 (87.9)	0.210 ^+^
	Yes	0 (0.0)	25 (17.5)	4 (12.1)	
Need for ICU stay	No	3 (23.1)	3 (2.1)	2 (6.1)	0.006 ^++^
	Yes	10 (76.9)	140 (97.9)	31 (93.9)	
Need for ICU stay (days), median (IQR)		0.5 (0–1)	1 (0–2)	0 (0–1)	<0.001 ^‡^

^+^ Pearson’s chi-square test ^++^ Fisher’s exact test ^‡^ Kruskal–Wallis test. Abbreviations: PAS, placenta accreta spectrum; IQR, interquartile range; ICU, intensive care unit.

**Table 4 diagnostics-14-00925-t004:** Participants’ outcomes associated with placental location.

		Location			*p*
		Anterior	Anterior/Posterior	Posterior	
		*N* (%)	*N* (%)	*N* (%)	
PAS (intraoperative/histology)	No	7 (13.0)	9 (10.2)	18 (43.9)	<0.001 ^+^
	Yes	47 (87.0)	79 (89.8)	23 (56.1)	
Degree of invasiveness (intraoperative/histology)	No	7 (13.0)	9 (10.2)	18 (43.9)	<0.001 ^+^
	Percreta	25 (46.3)	52 (59.1)	3 (7.3)	
	Accreta/increta	22 (40.7)	27 (30.7)	20 (48.8)	
Extrauterine spread (intraoperative)	No	37 (68.5)	41 (46.6)	40 (97.6)	<0.001 ^+^
	Yes	17 (31.5)	47 (53.4)	1 (2.4)	
Bladder involvement (intraoperative)	No	37 (68.5)	41 (46.6)	41 (100)	<0.001 ^+^
	Yes	17 (31.5)	47 (53.4)	0 (0)	
Parametrial involvement (intraoperative)	No	46 (85.2)	63 (71.6)	40 (97.6)	0.001 ^+^
	Yes	8 (14.8)	25 (28.4)	1 (2.4)	
Blood loss during surgery, median (IQR)		500 (0–1500)	1250 (500–2500)	250 (0–750)	<0.001 ^‡^
Need for massive transfusion	No	49 (90.7)	58 (69.0)	40 (100.0)	<0.001 ^+^
	Yes	5 (9.3)	26 (31.0)	0 (0.0)	
Hysterectomy	No	38 (70.4)	39 (44.3)	40 (97.6)	<0.001 ^+^
	Yes	16 (29.6)	49 (55.7)	1 (2.4)	
Bladder repair	No	37 (68.5)	41 (46.6)	41 (100.0)	<0.001 ^+^
	Yes	17 (31.5)	47 (53.4)	0 (0.0)	
Duration of delivery (min), median (IQR)		70 (50–100)	90 (60–145)	60 (50–70)	<0.001 ^‡^
Acute onset of labor	No	47 (87)	74 (84.1)	33 (80.5)	0.687 ^+^
	Yes	7 (13)	14 (15.9)	8 (19.5)	
Need for ICU stay	No	5 (9.3)	0 (0.0)	3 (7.3)	0.007 ^++^
	Yes	49 (90.7)	88 (100.0)	38 (92.7)	
Need for ICU stay (days), median (IQR)		1 (0–1)	1 (0–2)	0 (0–1)	<0.001 ^‡^

^+^ Pearson’s chi-square test ^++^ Fisher’s exact test ^‡^ Kruskal–Wallis test. Abbreviations: PAS, placenta accreta spectrum; IQR, interquartile range; ICU, intensive care unit

**Table 5 diagnostics-14-00925-t005:** Multiple regression analysis results with all under study outcomes as dependent variables and previa type as independent variable.

Dependent Variable	Independent Variable	OR (95% CI) ^+^	*p*
PAS intraoperative/histology)	No previa vs. low-lying/marginal	14.80 (1.30–168.93)	0.030
	Complete/partial vs. low-lying/marginal	3.32 (1.29–8.56)	0.013
Degree of invasiveness (intraoperative/histology) ^1^	No previa vs. low-lying/marginal	1.79 (0.32–9.93)	0.508
	Complete/partial vs. low-lying/marginal	0.69 (0.22–2.13)	0.521
Extrauterine spread (intraoperative)	No previa vs. low-lying/marginal	0.29 (0.02–5.22)	0.401
	Complete/partial vs. low-lying/marginal	4.06 (1.31–12.56)	0.015
Bladder involvement (intraoperative)	No previa vs. low-lying/marginal	0.27 (0.01–5.78)	0.403
	Complete/partial vs. low-lying/marginal	4.33 (1.35–13.87)	0.014
Parametrial involvement (intraoperative)	No previa vs. low-lying/marginal	-	- ^‡^
	Complete/partial vs. low-lying/marginal	8.88 (1.12–70.60)	0.039
Need for massive transfusion	No previa vs. low-lying/marginal	0.54 (0.01–46.06)	0.787
	Complete/partial vs. low-lying/marginal	6.75 (0.84–54.31)	0.073
Hysterectomy	No previa vs. low-lying/marginal	1.68 (0.27–10.37)	0.575
	Complete/partial vs. low-lying/marginal	3.73 (1.19–11.65)	0.024
Bladder repair	No previa vs. low-lying/marginal	0.27 (0.01–5.78)	0.403
	Complete/partial vs. low-lying/marginal	4.33 (1.35–13.87)	0.014
Acute onset of labor	No previa vs. low-lying/marginal	-	- ^‡^
	Complete/partial vs. low-lying/marginal	1.03 (0.29–3.68)	0.964
Need for ICU	No previa vs. low-lying/marginal	0.16 (0.02–1.42)	0.099
	Complete/partial vs. low-lying/marginal	2.77 (0.38–20.178)	0.315
		**β (SE) ^++^**	** *p* **
Blood loss during surgery	No previa vs. low-lying/marginal	−0.24 (0.15)	0.105
	Complete/partial vs. low-lying/marginal	0.17 (0.09)	0.056
Duration of delivery	No previa vs. low-lying/marginal	−0.01 (0.07)	0.867
	Complete/partial vs. low-lying/marginal	0.11 (0.04)	0.011
Need for ICU stay (days)	No previa vs. low-lying/marginal	0.03 (0.08)	0.669
	Complete/partial vs. low-lying/marginal	0.14 (0.04)	0.001

^+^ Odds Ratio (95% Confidence Interval) adjusted for age, prior uterine surgeries, maternal comorbidities, gestational age at delivery and number of c-section and fetuses; ^++^ Regression coefficient (Standard Error) adjusted for age, prior uterine surgeries, maternal comorbidities, gestational age at delivery, number of IVFs fetuses and number of c-section ^‡^ could not be computed due to no distribution ^1^ in cases with PAS (Accreta/increta vs. Percreta).

**Table 6 diagnostics-14-00925-t006:** Multiple regression analysis results with all under study outcomes as dependent variables and location as independent variable.

Dependent Variable	Independent Variable	OR (95% CI) ^+^	*p*
PAS (intraoperative/histology)	Anterior vs. posterior	6.94 (2.11–22.81)	0.001
	Anterior/posterior vs. posterior	7.38 (2.39–22.77)	0.001
Degree of invasiveness (intraoperative/histology) ^1^	Anterior vs. posterior	0.11 (0.02–0.59)	0.010
	Anterior/posterior vs. posterior	0.06 (0.01–0.32)	0.001
Extrauterine spread (intraoperative)	Anterior vs. posterior	19.27 (2.06–179.89)	0.009
	Anterior/posterior vs. posterior	49.63 (5.29–465.57)	0.001
Bladder involvement (intraoperative)	Anterior vs. posterior	-	- ^‡^
	Anterior/posterior vs. posterior	-	- ^‡^
Parametrial involvement (intraoperative)	Anterior vs. posterior	6.35 (0.71–56.62)	0.098
	Anterior/posterior vs. posterior	12.12 (1.42–103.17)	0.022
Need for massive transfusion	Anterior vs. posterior	-	- ^‡^
	Anterior/posterior vs. posterior	-	- ^‡^
Hysterectomy	Anterior vs. posterior	16.32 (1.70–157.18)	0.016
	Anterior/posterior vs. posterior	55.49 (5.73–537.27)	0.001
Bladder repair	Anterior vs. posterior	-	- ^‡^
	Anterior/posterior vs. posterior	-	- ^‡^
Acute onset of labor	Anterior vs. posterior	0.28 (0.08–1.07)	0.062
	Anterior/posterior vs. posterior	0.24 (0.07–0.92)	0.037
Need for ICU	Anterior vs. posterior	0.45 (0.07–2.84)	0.395
	Anterior/posterior vs. posterior	-	- ^‡^
		**β (SE) ^++^**	** *p* **
Blood loss during surgery	Anterior vs. posterior	0.15 (0.10)	0.126
	Anterior/posterior vs. posterior	0.27 (0.09)	0.003
Duration of delivery	Anterior vs. posterior	0.05 (0.05)	0.263
	Anterior/posterior vs. posterior	0.15 (0.04)	0.001
Need for ICU stay (days)	Anterior vs. posterior	0.08 (0.05)	0.099
	Anterior/posterior vs. posterior	0.16 (0.05)	<0.001

^+^ Odds Ratio (95% Confidence Interval) adjusted for age, prior uterine surgeries, maternal, comorbidities, gestational age at delivery, number of c-section and number of fetuses; ^++^ Regression, coefficient (Standard Error) adjusted for age, prior uterine surgeries, maternal comorbidities, gestational age at delivery, number of IVFs, number of c-section and number of fetuses ^‡^ could not, be computed due to no distribution ^1^ in cases with invasive placenta (accreta/increta vs. percreta), Abbreviations: PAS, placenta accrete spectrum; ICU, intensive care unit.

## Data Availability

All imaging data were provided by the 1st Department of Radiology, School of Medicine, National and Kapodistrian University of Athens, Aretaieion Hospital, Athens, Greece. These data can be available upon request and they are not fully uploaded to publicly accessible links due to the General Data Protection Regulation (GDPR) policy of the Hospital.

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
