# Peer review of "MRI Assessed Placental Location as a Diagnostic Tool of Placental Invasiveness and Maternal Peripartum Morbidity"

_diagnostics, 2024, doi:10.3390/diagnostics14090925_

Round 1
Reviewer 1 Report
Comments and Suggestions for Authors
The authors' intention was to investigate any association of the presence and type of placenta previa on MRI with PAS and maternal peripartum outcome.
They found that posterior placental location was associated with lower rates of PAS and better clinical outcomes compared to those placentas which involve the anterior uterine wall.
The paper is well written an appropriately support their findings.
I have only two comments:
1. The findings of this study are important but previously already published (reference 4,13,14). So, this paper just confirms already published data. They have to admit it in their discussion.
2. The suggestion that MRI should be done in all PAS cases is good but not achievable in many parts of the world. It is expensive and the availability is limited. I suggest that the authors present an algorithm in the manuscript how to screen and identify these cases and just the very high-risk cases should be sent for MRI. It would be a new message in this paper.
Author Response
I have only two comments:
Reviewer comment#1: The findings of this study are important but previously already published (reference 4,13,14). So, this paper just confirms already published data. They have to admit it in their discussion.
Response to comment #1: We would like to thank the reviewer for his/her kind comment. Indeed, our results are in accordance with those published in the literature. However, unlike the other studies, we have used MRI, which we believe provides more objective and reproducible information compared to ultrasound, to identify the previa type. We now comment on this information in the discussion section of the revised manuscript (page 15, lines 366-369).
Reviewer comment#2: The suggestion that MRI should be done in all PAS cases is good but not achievable in many parts of the world. It is expensive and the availability is limited. I suggest that the authors present an algorithm in the manuscript how to screen and identify these cases and just the very high-risk cases should be sent for MRI. It would be a new message in this paper.
Response to comment#2: We agree with the reviewer’s comment. Indeed, despite its added value to PAS diagnosis, it is difficult to perform MRI in all patients at high risk for PAS, especially in median-low-income countries due to cost effectiveness issues and limited availability. In our tertiary center, we perform MRI in all patients with complete previa on second trimester ultrasound and pertinent clinical history (i.e. history of c-section, in vitro fertilization, multiple uterine interventions or history of previous PAS pregnancy), even in cases with normal sonographic findings. This information is now clearly stated in the revised manuscript (page 15, lines 400-406).
Reviewer 2 Report
Comments and Suggestions for Authors
Dear Authors,
The presented study tackles the issue of the predictive role of MRI in placental invasiveness and prediction of maternal peripartum morbidity. The study was conducted reliably with an appropriate selection of tests. Overall, I think that this article should be published, however, some issues require complementary information:
1. I suggest changing the title to “MRI Assessed Placental Location and Invasiveness as Diagnostic Tool of Maternal Peripartum Morbidity.”
2. The abstract should be without headings.
3. The first sentence in the Abstract should be the background, not the aim.
4. The reference type is not correct. Please follow the author's guidelines.
5. It’s worth adding in the Introduction available diagnostic methods of PAS syndrome and their sensitivity and specificity.
6. I suggest adding the number of ethics committee approval and the statement that the study was performed according to the Helsinki Declaration (Please see instructions for authors)
7. I suggest including only 3 of the most significant figures.
8. It is worth dividing Table 1 into two tables (Table 1- sociodemographic and medical characteristics, Table- 2 Placenta related findings) and elaborating a little on the sociodemographic findings and then always mark it in the text as a reference e.g. (Table 1).
9. It is worth adding information about the validity of MRI in PAS syndrome diagnosis. Did you check if your findings from the MRI were in line with intraoperative findings (especially placental invasiveness)? In my opinion, it should be the major outcome of this study.
10. It is worth adding more and updated references in this field.
Author Response
Dear Authors,
The presented study tackles the issue of the predictive role of MRI in placental invasiveness and prediction of maternal peripartum morbidity. The study was conducted reliably with an appropriate selection of tests. Overall, I think that this article should be published, however, some issues require complementary information:
Reviewer comment#1: I suggest changing the title to “MRI Assessed Placental Location and Invasiveness as Diagnostic Tool of Maternal Peripartum Morbidity.”
Response to comment#1: We would like to thank the reviewer for his/her comment. According to the reviewer’s suggestion, we have modified the title of our manuscript to “MRI Assessed Placental Location as a diagnostic tool of Placental Invasiveness and Maternal Peripartum Morbidity” (page 1, lines 2-3).
Reviewer comment#2: abstract should be without headings.
Response to comment#2: As suggested by the reviewer, we have removed the abstract headings in the revised manuscript (page 1, lines 27-45)
Reviewer comment#3: The first sentence in the Abstract should be the background, not the aim.
Response to comment#3: We agree with the reviewer’s comment, and we have added an appropriate abstract background sentence in the revised manuscript (page 1, lines 27-29).
Reviewer comment#4: The reference type is not correct. Please follow the author's guidelines.
Response to comment#4: According to the reviewer’s suggestion and authors’ guidelines, we have used the ACS style for the reference list (see Reference section of our revised manuscript).
Reviewer comment#5: It’s worth adding in the Introduction available diagnostic methods of PAS syndrome and their sensitivity and specificity.
Response to comment#5: We agree with the reviewer’s comment and we have included information regarding the diagnostic accuracy of US and MRI for PAS detection in the introduction section (page 2, lines 68-72)
Reviewer comment#6: I suggest adding the number of ethics committee approval and the statement that the study was performed according to the Helsinki Declaration (Please see instructions for authors)
Response to comment#6: Following the reviewer’s suggestion we have included the above information in our revised manuscript (Page 2, lines 82-83). This information is also presented at the end of the manuscript (see Institutional Review Board Statement, page 16, lines 437-439).
Reviewer comment#7: I suggest including only 3 of the most significant figures.
Response to comment#7: This study focuses on the diagnostic ability of MRI for placenta previa and PAS identification in the pregnant population; we believe that the provided MRI Figures strengthen the content of the manuscript and we, therefore, decided not to omit any of them. However, in the revised manuscript we have grouped the first four figures together as Figure 1, to better demonstrate the differences among the different types of previa on MRI and limit the overall number of individual Figures (page 4). We have revised the corresponding Figure legend accordingly.
Reviewer comment#8: It is worth dividing Table 1 into two tables (Table 1- sociodemographic and medical characteristics, Table- 2 Placenta related findings) and elaborating a little on the sociodemographic findings and then always mark it in the text as a reference e.g. (Table 1).
Response to comment#8: We agree with the reviewer’s comment, and we have divided Table 1 into two separate Tables in the revised manuscript (Table 1 and Table 2 in the revised manuscript, page 7-9).
Reviewer comment#9: It is worth adding information about the validity of MRI in PAS syndrome diagnosis. Did you check if your findings from the MRI were in line with intraoperative findings (especially placental invasiveness)? In my opinion, it should be the major outcome of this study.
Response to comment#9: Following the reviewer’s suggestion we have included more information on the validity of MRI in PAS diagnosis in the revised manuscript (page 15, lines 393-399). The overall diagnostic ability of MRI in PAS diagnosis in our study population using the combination of several MRI features was high for both 1.5T and 3.0T groups. The standard of reference was the intraoperative information and, in cases of hysterectomy, the pathological information as well. In particular, MRI sensitivity and specificity for PAS were 95.7% (95% CI: 87.8−99.1) and 81.8% (95% CI: 59.7−94.8) for 1.5-T and 93.8% (95% CI: 86.0−97.9) and 81.8% (95% CI: 48.2−97.7) for 3.0-T, with no significant differences in sensitivity and specificity between both groups (P = 0.725and P > 0.999, respectively). Sensitivity and specificity for MRI detection of extrauterine placental spread were 100% (95% CI: 89.4−100.0) and 96.7% (95% CI: 88.1−99.6) for 1.5-T and 97% (95% CI: 84.2−99.9) and 96.6% (95% CI: 88.1−99.6) for 3.0-T; no statistically significant differences were found between groups (P > 0.999, for both comparisons).The above results have already been reported in a previous publication (Bourgioti C, Zafeiropoulou K, Tzavara C, Daskalakis G, Fotopoulos S, Theodora M, Nikolaidou ME, Konidari M, Gourtsoyianni S, Panourgias E, Koutoulidis V, Martzoukos EA, Konstantinidou AE, Moulopoulos LA. Comparison between 1.5-T and 3.0-T MRI for the diagnosis of placenta accreta spectrum disorders. Diagn Interv Imaging. 2022 Sep;103(9):408-417. doi: 10.1016/j.diii.2022.04.005). The appropriate reference is now included in the revised manuscript (Ref # 23, in the revised manuscript).
Reviewer comment#10: It is worth adding more and updated references in this field.
Response to comment#10: According to the reviewer’s suggestion we have increased the number of references in our revised manuscript (Ref # 6,7,8,21,22,23)